

# Effects of rheumatoid arthritis associated transcriptional changes on osteoclast differentiation network in the synovium

Shilpa Harshan[1,*], Poulami Dey[1,2,*] and Srivatsan Ragunathan[1]

[1] Institute of Bioinformatics and Applied Biotechnology, Bangalore, Karnataka, India
[2] Manipal Academy of Higher Education, Manipal, Karnataka, India
* These authors contributed equally to this work.

## ABSTRACT

**Background:** Osteoclast differentiation in the inflamed synovium of rheumatoid arthritis (RA) affected joints leads to the formation of bone lesions. Reconstruction and analysis of protein interaction networks underlying specific disease phenotypes are essential for designing therapeutic interventions. In this study, we have created a network that captures signal flow leading to osteoclast differentiation. Based on transcriptome analysis, we have indicated the potential mechanisms responsible for the phenotype in the RA affected synovium.

**Method:** We collected information on gene expression, pathways and protein interactions related to RA from literature and databases namely Gene Expression Omnibus, Kyoto Encyclopedia of Genes and Genomes pathway and STRING. Based on these information, we created a network for the differentiation of osteoclasts. We identified the differentially regulated network genes and reported the signaling that are responsible for the process in the RA affected synovium.

**Result:** Our network reveals the mechanisms underlying the activation of the neutrophil cytosolic factor complex in connection to osteoclastogenesis in RA. Additionally, the study reports the predominance of the canonical pathway of NF-κB activation in the diseased synovium. The network also confirms that the upregulation of T cell receptor signaling and downregulation of transforming growth factor beta signaling pathway favor osteoclastogenesis in RA. To the best of our knowledge, this is the first comprehensive protein–protein interaction network describing RA driven osteoclastogenesis in the synovium.

**Discussion:** This study provides information that can be used to build models of the signal flow involved in the process of osteoclast differentiation. The models can further be used to design therapies to ameliorate bone destruction in the RA affected joints.

Corresponding author
Srivatsan Ragunathan,
srivatsan@ibab.ac.in

# INTRODUCTION

Rheumatoid arthritis (RA) is a systemic autoimmune disease that primarily affects synovial joints. The disease is characterized by chronic inflammation in the joints, leading
to synovial hyperplasia (pannus formation), destruction of the cartilage and erosion of the underlying bone. RA is a complex disease involving several molecular pathways across various cell types and tissues. Thus in order to elucidate the underlying cause of a particular phenotype associated to the disease, identification of the network consisting of differentially expressed genes (DEGs) in the interacting pathways is essential. Studies have used pathway analysis to identify affected pathways from lists of DEGs (*Hao et al., 2017*; *Wang et al., 2017*; *Lee et al., 2011*; *Wu et al., 2010*). The lists have also been used to create networks that are related to specific diseases or conditions. Earlier work using RA samples has focused on generating networks of the genes showing differential regulation (*Hao et al., 2017*; *Wang et al., 2017*) or the most enriched gene ontology (GO) (*The Gene Ontology Consortium, 2017*) category in the DEG lists (*Lee et al., 2011*). A comprehensive network describing molecular interactions across various RA affected tissues was created using publicly available microarray data by *Wu et al. (2010)*. Other groups have created gene regulatory networks (GRNs) using in vitro data from cultured fibroblasts and macrophages (*Kupfer et al., 2014*; *You et al., 2014*).

*Kupfer et al. (2014)* used time series data generated from RA synovial fibroblasts subjected to external stimulation to create a GRN. They simulated the network to analyze the behavior of genes involved in RA pathogenesis, in response to stimulation by RA associated cytokines and growth factors (GFs). *You et al. (2014)* created a GRN and identified the critical interactions responsible for synovial fibroblast invasiveness in RA synovium. The creation of a detailed protein–protein interaction (PPI) network describing the connections between various pathways involved in any specific RA process, at the level of the synovial tissue, is yet to be attempted. In this study, using the publicly available gene expression data for RA synovial tissue and protein interactions and pathway databases, we created and analyzed a detailed phenotype-specific PPI network. We used differentially regulated genes to identify the altered pathways in the affected synovium. We identified the pathway of osteoclast differentiation as a phenotype connected to many of the altered pathways in the RA synovium. It is established that the RA synovium harbors osteoclasts, the cells responsible for bone degradation in the affected joints (*Schett, 2007*). Therefore, a network of proteins participating in the interacting pathways underlying the RA associated process of osteoclast differentiation in the synovium was created for the first time. We report the upregulated signaling routes that drive osteoclastogenesis via the generation of reactive oxygen species (ROS) by neutrophil cytosolic factor (NCF) complex in the RA synovium. We demonstrate the contribution of elevated T cell receptor signaling in facilitating osteoclast differentiation in the affected tissue. In addition, we describe the importance of the canonical pathway of NF-κB activation and the transforming growth factor beta (TGFβ) pathway in connection to the process. Finally, the network reports all the possible routes by which the inflamed synovium promotes the differentiation of osteoclasts.

## MATERIALS AND METHODS

This study involved two major steps: selection of a phenotype exhibited by the RA synovium, and construction and analysis of a PPI network for the selected

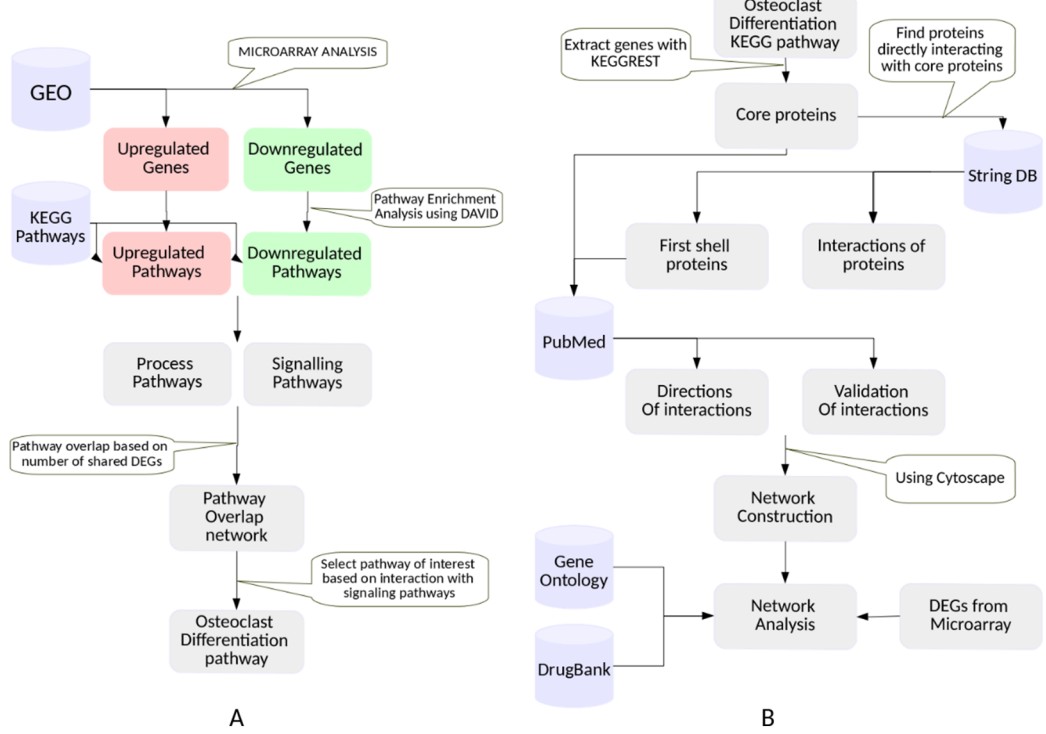

**Figure 1 The workflow.** (A) Selection of a KEGG pathway representing a phenotype exhibited by the RA synovium. The differentially expressed genes in the RA synovium were identified from the analysis of the microarray datasets obtained from GEO database. A KEGG pathway enrichment analysis was performed on the differentially expressed gene lists. The enriched pathways were categorized into process and signaling pathways. Based on the shared differentially expressed genes, a pathway overlap network was created. Osteoclast differentiation pathway was selected as the pathway of interest as it overlapped with most number of signaling pathways. (B) Construction and analysis of the osteoclast differentiation network. The proteins belonging to the KEGG osteoclast differentiation pathway were termed as "core proteins." The core proteins were used as an input to the String DB to obtain the proteins interacting with them (first shell proteins). The interactions among all the proteins (core and first shell) were also extracted from String DB. The interactions were validated using PubMed. The directions of the protein interactions were also obtained. The proteins and the interactions were used to construct a network for osteoclast differentiation. The gene ontology term enrichment analysis was performed on the network. Differentially regulated genes were indicated in the network. The gene ontology term enrichment analysis and the differentially regulated genes were used to identify the important protein interactions that lead to osteoclast differentiation in the RA synovium. The protein targets of the drugs used in RA treatment were obtained from DrugBank database and were indicated in the network.

phenotype. Figure 1 shows the detailed workflow that was followed. Each step is described in detail in this section. The databases used in this study are summarized in Table 1.

## Identification of DEGs using microarray data analysis

The DEGs were obtained by re-analyzing the publicly available microarray datasets in Gene Expression Omnibus (GEO) (*Edgar, Domrachev & Lash, 2002*) database. The repository was searched for the data generated from synovial tissue in RA patients and

**Table 1 The details of the databases used in this study.**

| Database | Type of data obtained for this study | Features | Rationale |
|---|---|---|---|
| GEO (*Edgar, Domrachev & Lash, 2002*) | Microarray gene expression data | GEO is a public repository with easy access to high throughput data, including microarray data and related metadata such as tissue type, disease state, etc. | Microarray data from GEO database was used to identify DEGs in the RA synovium |
| DAVID—Gene ID conversion tool (*Huang, Sherman & Lempicki, 2009b, 2009a*) | Gene ID types | The DAVID knowledge base supports conversion between more than 20 gene ID types, including Affymetrix probe IDs | The DAVID gene ID conversion table was used to convert Affymetrix probe IDs to Entrez IDs |
| KEGG pathway (*Kanehisa et al., 2017*) | Molecular pathways | KEGG pathways are manually drawn and frequently updated. References are provided for each pathway | The KEGG pathway database was used to identify the enriched pathways from the list of DEGs |
| String (v10) (*Szklarczyk et al., 2015*) | Protein–protein interaction | String integrates information from several sources including experimental results from literature, and provides a confidence score for the interaction | The String database was used to identify the proteins that directly interacted with KEGG osteoclast differentiation pathway proteins. These proteins (KEGG osteoclast differentiation pathway proteins and their interactors) were used to create a PPI network for osteoclast differentiation |
| Gene Ontology (*The Gene Ontology Consortium, 2017*) | Functional annotation | Gene Ontology annotation associates genes to specific functional terms. A GO enrichment analysis provides information about functions that a set of genes may be involved in | GO enrichment was used to identify the functions associated with the proteins in the osteoclast differentiation PPI network |
| DrugBank (*Wishart et al., 2018*) | Drug targets | DrugBank provides comprehensive information of drug targets and drug types (small molecule, biologics, etc.) | DrugBank was used to identify proteins in the network that are targets of RA drugs |

**Note:**
The name and the reference of the database is listed in the column named "Database." The type of data used from the database and the features of the database are described in the table. The rationale for using the data from each of the databases in our study is included in the table.

**Table 2 The details of the microarray datasets from GEO repository used in our study.**

| Accession Number | Platform | Rheumatoid arthritis samples | Healthy control samples |
|---|---|---|---|
| GSE1919 (*Ungethuem et al., 2010*) | Affymetrix Human Genome U95A Array | 5 | 5 |
| GSE7307 | Affymetrix Human Genome U133 Plus 2.0 Array | 5 | 9 |
| GSE12021 (*Huber et al., 2008*) | Affymetrix Human Genome U133A Array | 12 (3 m/9 f) | 9 (7 m/2 f) |
| | Affymetrix Human Genome U133B Array | 12 (3 m/9 f) | 4 (3 m/1 f) |
| GSE55235 (*Woetzel et al., 2014*) | Affymetrix Human Genome U133A Array | 10 | 10 |
| GSE55457 (*Woetzel et al., 2014*) | Affymetrix Human Genome U133A Array | 13 | 10 |
| GSE77298 (*Broeren et al., 2016*) | Affymetrix Human Genome U133 Plus 2.0 Array | 16 | 7 |

**Note:**
The accession number and the reference of the datasets is listed in the column named "Accession Number." The title of the Affymetrix platform used by the particular dataset in included in the "Platform" column of the table. The information on the number of the RA patients and healthy controls used in each dataset are included in this table. Gender distribution is provided for the datasets GSE12021 (U133A) and 12021 (U133B).

healthy controls. The results were further narrowed down by considering only the data from Affymetrix platforms with at least four RA and four control samples. Datasets selected for the study are mentioned in Table 2.

Of the seven datasets, information regarding treatments received by the patients was not available for GSE77298 (*Broeren et al., 2016*) and GSE7307. Earlier, it was established that the differential regulation of the genes in these datasets was not under the influence of drug therapy (*Dey, Panga & Raghunathan, 2016*). The clinical information for the RA patients was available for the datasets GSE1919 (*Ungethuem et al., 2010*), GSE12021 (U133A) (*Huber et al., 2008*), GSE12021 (U133B) (*Huber et al., 2008*), GSE55235 (*Woetzel et al., 2014*) and GSE55457 (*Woetzel et al., 2014*). The erythrocyte sedimentation rate (ESR) and the concentration of C-reactive protein (CRP) reported for these datasets were higher than 40 mm $h^{-1}$ and 21 mg $l^{-1}$, respectively. These values indicated active inflammation in the synovium of the RA patients (*Wetteland et al., 1996*; *Otterness, 1994*). For the datasets, GSE77298 and GSE7307, the values for these parameters were not available.

Raw data from the seven datasets along with their metadata was downloaded using the R libraries GEOquery (*Davis & Meltzer, 2007*) and GEOmetadb (*Zhu et al., 2008*). The data was analyzed using the affy (*Gautier et al., 2004*) and simpleaffy (*Wilson & Miller, 2005*) libraries in the Bioconductor package in R (*R Core Team, 2017*).

In this analysis, two algorithms, Robust Multi-array Average (RMA) and Microarray Suite 5.0 (MAS5) were used for the data normalization. The choice of data normalization algorithms affects the final selection of the DEGs (*Pepper et al., 2007*). In order to reduce the algorithm specific effects, both RMA and MAS5 were used in this study. In the case of MAS5, probesets having at least one present call ("P") in control as well as treatment samples were considered. Probesets were annotated with Entrez Identifiers (IDs) using the Bioconductor as well as Database for Annotation, Visualization and Integrated Discovery (DAVID) gene-ID conversion tool (*Huang, Sherman & Lempicki, 2009b*; *Huang, Sherman & Lempicki, 2009a*). Welch *t* test was applied to calculate the significance for differential expression between the RA and the control samples. As per the recommendations by *Huang, Sherman & Lempicki (2009a)*, in our study a gene with a linear fold change of 2 (for up- and downregulation) and a *p* value $\leq 0.05$ was considered to be differentially expressed. A final list of DEGs, from the seven datasets, was obtained using the selection rules as described below:

A) In a dataset, a gene is considered to be upregulated if:

   i)  It is upregulated in both RMA and MAS5 [+, +]
       Or,
   ii)  It is upregulated in one of the algorithm and not differentially expressed in the other [+, 0] or [0, +].

B) Across the datasets, the gene is upregulated if:

   i)  It is [+, +] in at least one of the seven datasets and no downregulation in any of the datasets
       Or,
   ii)  It is [+, 0], in at least one dataset and [0, +] in at least one of the remaining datasets while there is no downregulation in any one of them.

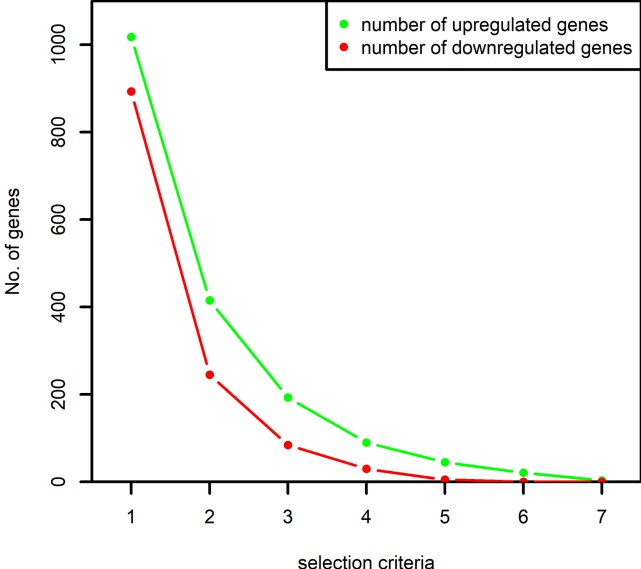

**Figure 2 Number of up and downregulated genes obtained based on the selection criteria.** The numbers on *x*-axis represent the following: "1": selection of the genes at least once by the algorithm MAS5 and RMA, "2": selection of the genes at least twice by both the algorithms, "3": selection of the genes at least thrice by both the algorithms and so on. The *y*-axis represents the number of selected up and downregulated genes. The number of differentially expressed genes selected at least once by both the algorithms are used in this study.           

The same procedure was repeated for the downregulated genes.

When the selection criteria was made more stringent by demanding the selection of a gene in at least two datasets, the number of selected genes reduced by almost 50% as shown in Fig. 2. Since the aim of the study was to identify all the important signaling associated with the RA-associated process we decided to proceed with the selection criteria of presence in at least one dataset.

Finally, we prepared a list of up and downregulated genes which we named as "common-up" and "common-down," respectively.

## Pathway analysis

The common-up and common-down gene lists were separately examined for the enrichment of pathways listed in the Kyoto Encyclopedia of Genes and Genomes (KEGG) database (*Kanehisa et al., 2017*) using DAVID. For the enrichment analysis, we created a custom background by combining the total probesets present on all four microarray platforms and annotating them with Entrez IDs using the DAVID gene ID conversion tool. The pathways which were significantly over-represented in the common-up or common-down gene lists with an Expression Analysis Systematic Explorer (EASE) score ≤0.05 and fold enrichment ≥1.5 were considered to be affected in RA (*Huang, Sherman & Lempicki, 2009a*). EASE score is a modified one-tailed Fisher exact probability used in enrichment analysis (*Hosack et al., 2003*).

The pathways were grouped according to their KEGG categories. Those belonging to the categories "human diseases," "metabolism" or the ones lacking PPIs were not considered further. The category "human diseases" contains pathways that represent

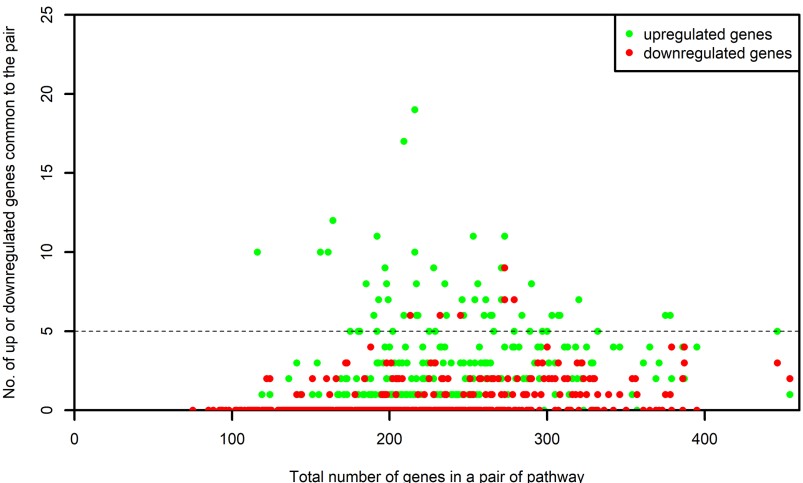

**Figure 3 Number of up and downregulated genes shared by pairs of pathways.** The numbers on *x*-axis represent the sum of all the genes belonging to the pathway pairs. The *y*-axis represents the number of up and downregulated genes common to the pairs. The pathway pairs sharing at least five up or five downregulated genes are used in this study.

specific disease conditions. These were excluded because they do not reflect the PPI in healthy conditions. "Metabolism" contains pathways that describe interconversions of metabolites. Since this study focused on PPI involved in the RA affected synovium, we did not consider the metabolic pathways. The remaining pathways were categorized based on their functional specificity. The pathways which result in a specific function like platelet activation were tagged as process pathways, whereas the pathways describing more general signaling events like the activation of multiple transcription factors (TFs) through T cell receptor signaling, were considered as signaling pathways.

The list of genes present in each selected pathway was downloaded from KEGG using the KEGGREST (*Tenenbaum, 2017*) package in R. For each one of the selected pathway, the list of genes that were common between the pathway and the microarray platforms was created. We name this as "S-list." By the pairwise intersection of the S-list of each process pathway with every one of the non-disease pathway, we obtained the DEGs shared between the pairs of process and non-disease pathways. The number of DEGs shared by each pair was examined. The pairs of pathways sharing at least five upregulated or five downregulated genes were retained for our study. Figure 3 shows the number of up or downregulated genes common to the pair of pathways. It is evident from this figure that the above mentioned criterion did not result in a bias toward pairs of larger pathways.

## Construction of network

We constructed a PPI network for the osteoclast differentiation. The osteoclast differentiation pathway (ODP) proteins, obtained from KEGG were defined as a set of core proteins. Interactors of the core proteins (first-shell interactors) were extracted from the STRING database (version 10) (*Szklarczyk et al., 2015*). The protein list was restricted by considering only experimentally validated interactions with a score of $\geq 0.9$. This score on a scale of 0–1 represents the confidence of experimental validation with

maximum confidence being 1. We obtained the directions for these interactions from the literature references used in STRING, when available, or with a separate literature search in PubMed. The complete network was built in Cytoscape (*Shannon et al., 2003*) using all the obtained interactions. The proteins corresponding to the DEGs in RA synovium obtained from the microarray data analysis were indicated in this network.

## Analysis of the network

The network was a mixed network consisting of the undirected protein binding edges and the directed edges of activation, inhibition or the post-translational modification (PTM). The nodes are labeled using the official gene symbols corresponding to the proteins used to create the network. The interactions involved undirected PPIs or directional PTM like phosphorylation, methylation, acetylation, ubiquitination, etc. We included activation or inhibition as an interaction whenever the reference mentioned that the target protein is activated or inhibited as a result of the interaction. We created a version of this directed network without ubiquitin C and its edges. We named this version as "directed ODP network."

We conducted GO enrichment analysis on the directed ODP network proteins using the GO molecular function (GOMF) and GO biological process (GOBP) terms. Terms with an EASE score ≤0.05 and fold enrichment ≥1.5 were considered as enriched. We combined 23 enriched GOMF terms to identify the proteins that bind to DNA. In the case of the GOBP terms, we selected the enriched signaling terms that contained differentially regulated genes for T cell receptor signaling, B cell receptor signaling and FC-ε receptor signaling. For NF-κB signaling, toll-like receptor (TLR) signaling and TGFβ signaling pathways we combined six, nine and three enriched terms, respectively, and examined their differential regulation. Using the selected enriched signaling terms, we extracted subnetworks corresponding to each signaling pathway from the directed ODP network. All the subnetworks demonstrate the flow of information from the first-shell interactor proteins to the core proteins of the ODP network. The details of the GOMF and GOBP signaling terms enriched in the analysis is provided in the Tables S1 and S2, respectively.

The database DrugBank (*Wishart et al., 2018*) was explored to locate the target proteins of drugs that are commonly used in the treatment of RA. We used the information to pinpoint the network proteins which are targets of the RA drugs. The details regarding the drugs and their targets are submitted in Table S3.

We converted all the edges of the directed ODP network to single undirected protein binding edges to create an "undirected ODP network." We analyzed this network using the Cytoscape plugin NetworkAnalyzer (*Assenov et al., 2008*). We used the plugin MCODE v1.5.1. (*Bader & Hogue, 2003*) to identify the clusters in the network.

## RESULTS

### Differentially expressed genes in RA synovium

We analyzed seven Affymetrix microarray datasets from five different platforms. Out of 21,246 Entrez annotated genes measured, 1,018 upregulated and 893 downregulated genes were identified in the RA synovium compared to the healthy controls.

**Table 3 Datasets showing downregulation of AP1 proteins in the RA synovium.**

| Dataset | FOSB | JUN | JUNB |
| --- | --- | --- | --- |
| GSE1919 | Downregulation | No differential regulation | No differential regulation |
| GSE7307 | No differential regulation | Downregulation | No differential regulation |
| GSE12021 (U133A) | Downregulation | Downregulation | Downregulation |
| GSE55235 | Downregulation | Downregulation | Downregulation |
| GSE55457 | Downregulation | Downregulation | Downregulation |

Note:
This table includes the information about the downregulation of the AP1 proteins as obtained by different microarray datasets used in our study.

The differentially regulated genes are submitted in Table S4. Only three genes signal transducer and activator of transcription 1 (STAT1), interleukin 7 receptor and immunoglobulin kappa constant region (IGKC) were upregulated in all seven datasets. Interestingly, three activator protein 1 (AP1) proteins FosB proto-oncogene, AP-1 transcription factor subunit (FosB), Jun proto-oncogene, AP-1 transcription factor subunit (JUN) and JunB proto-oncogene, AP-1 transcription factor subunit (JUNB) were downregulated. Table 3 shows the datasets in which the genes were downregulated.

## Diverse pathways are involved in the disease processes affecting the RA synovium

Using the DEGs from the microarray analysis as the input, we found that 52 KEGG pathways were enriched in the upregulated gene list, and 29 in the downregulated gene list. The EASE scores of the selected pathways were much less that the cut-off of 0.05. The enrichment analysis when performed with the combined DEG list (1,018 up genes + 893 down genes) resulted in only 55 pathways. Among these, only two pathways were newly obtained when compared to the previous list of 52 upregulated and 29 downregulated pathways. As the combined analysis proved less informative, the 52 up and 29 downregulated pathways were considered for the study. Three pathways, namely, extra cellular matrix-receptor interaction, Focal adhesion and Proteoglycans in cancer occurred in both the up and downregulated pathway lists because each pathway had a significant number of up and downregulated genes. The KEGG category-wise distributions of the enriched pathways are shown in Figs. 4 and 5 and the detailed results of the pathway analysis are given in the Tables S5 and S6. A total of 26 of the upregulated pathways and four of the downregulated pathways belonged to the KEGG category "human diseases."

Among the upregulated non-disease pathways, the category "immune system" had the highest number of enriched pathways. Most of the immune receptor signaling pathways in this category were upregulated. Among the other signaling pathways, NF-κB and JAK-STAT signaling pathways were upregulated in our analysis. All the signaling pathways which belong to the KEGG categories, "immune system" and "signal transduction" that were enriched in the upregulated gene list are shown in Table 4.

Specialized cells called osteoclasts which facilitate bone resorption are also present in the invading pannus of the RA joints (*Gravallese et al., 1998*; *Jung et al., 2014*; *Nevius, Gomes & Pereira, 2016*). All the five microarray datasets which provided

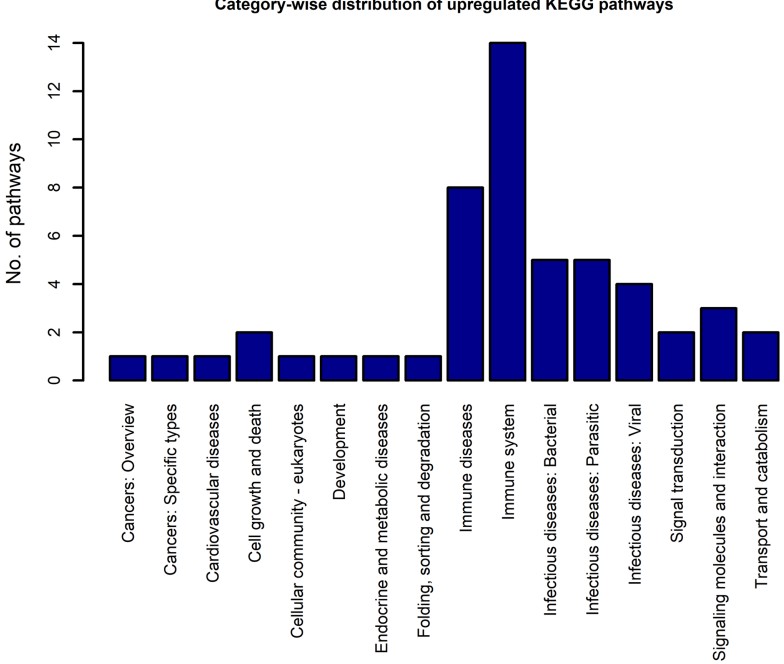

**Figure 4 The KEGG category-wise distribution of the enriched pathways in the upregulated genes of the RA synovium.** The *x*-axis represents the KEGG categories as listed in the KEGG database whereas the *y*-axis denotes the number of enriched pathways from our study belonging to the KEGG categories.

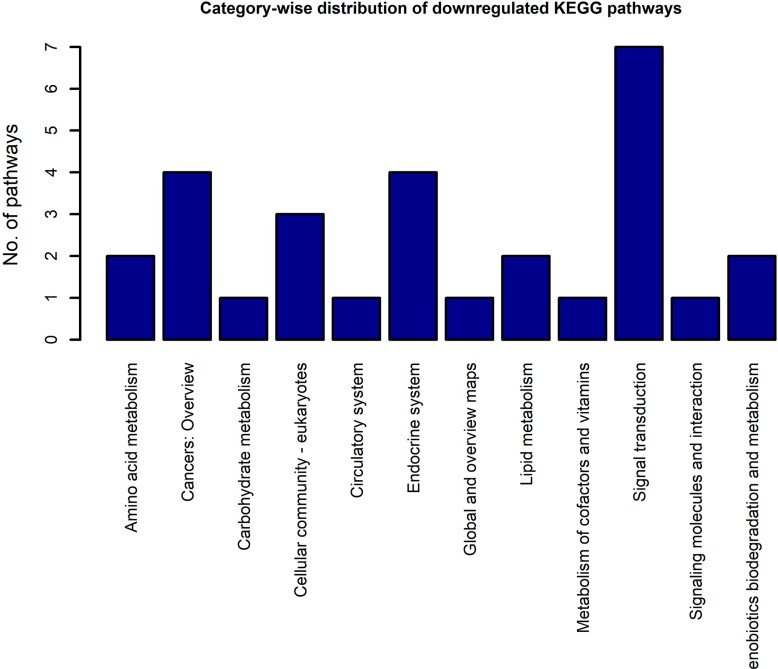

**Figure 5 The KEGG category-wise distribution of the enriched pathways in the downregulated genes of the RA synovium.** The *x*-axis represents the KEGG categories as listed in the KEGG database whereas the *y*-axis denotes the number of enriched pathways from our study belonging to the KEGG categories.

**Table 4 Upregulated signaling pathways in the RA synovium and their KEGG categories.**

| Pathways | KEGG categories | Fold enrichment | EASE score |
|---|---|---|---|
| hsa04064:NF-κB signaling pathway | Signal transduction | 3.92 | $6.25\ e^{-9}$ |
| hsa04664:Fc-ε RI signaling pathway | Immune system | 3.19 | $8.96\ e^{-5}$ |
| hsa04621:NOD-like receptor signaling pathway | Immune system | 3.11 | $6.68\ e^{-4}$ |
| hsa04662:B cell receptor signaling pathway | Immune system | 3.05 | $1.53\ e^{-4}$ |
| hsa04620:Toll-like receptor signaling pathway | Immune system | 2.89 | $1.35\ e^{-5}$ |
| hsa04062:Chemokine signaling pathway | Immune system | 2.82 | $4.96\ e^{-9}$ |
| hsa04660:T cell receptor signaling pathway | Immune system | 2.30 | $1.69\ e^{-3}$ |
| hsa04630:JAK-STAT signaling pathway | Signal transduction | 1.96 | $4.45\ e^{-3}$ |

Note:
The numbers prefixed with "hsa" are the KEGG identifiers for each pathway. The KEGG pathway term name along with the KEGG pathway identifier is listed in the column named "Pathways." The "Fold enrichment" is calculated based on the number of upregulated genes which belong to the KEGG pathway term ("Count"), total number of upregulated genes which are part of the KEGG pathway database ("List Total"), total number of genes that belong to the KEGG Pathway database ("Population Total") and number of the KEGG Pathway database genes which are part of the KEGG pathway term ("Population Hits"). Precisely, the "Fold enrichment" is the proportion of "Count"/"List Total" to "Population Hits"/"Population Total." The "EASE score" represents the significance of obtaining "Count" genes in the "List Total" for the given KEGG pathway term which has "Population Hits" genes in the background "Population Total."

**Table 5 Downregulated signaling pathways in the RA synovium and their KEGG categories.**

| Pathways | KEGG categories | Fold enrichment | EASE score |
|---|---|---|---|
| hsa04350:TGF-beta signaling pathway | Signal transduction | 2.42 | $9.49\ e^{-3}$ |
| hsa04152:AMPK signaling pathway | Signal transduction | 2.30 | $2.56\ e^{-3}$ |
| hsa04068:FoxO signaling pathway | Signal transduction | 2.09 | $6.57\ e^{-3}$ |
| hsa04022:cGMP-PKG signaling pathway | Signal transduction | 1.90 | $9.85\ e^{-3}$ |
| hsa04310:Wnt signaling pathway | Signal transduction | 1.83 | $3.26\ e^{-2}$ |
| hsa04024:cAMP signaling pathway | Signal transduction | 1.62 | $4.37\ e^{-2}$ |
| hsa04010:MAPK signaling pathway | Signal transduction | 1.55 | $3.27\ e^{-2}$ |

Note:
The numbers prefixed with "hsa" are the KEGG identifiers for each pathway. The KEGG pathway term name along with the KEGG pathway identifier is listed in the column named "Pathways." The "Fold enrichment" is calculated based on the number of downregulated genes which belong to the KEGG pathway term ("Count"), total number of downregulated genes which are part of the KEGG pathway database ("List Total"), total number of genes that belong to the KEGG Pathway database ("Population Total") and number of the KEGG Pathway database genes which are part of the KEGG pathway term ("Population Hits"). Precisely, the "Fold enrichment" is the proportion of "Count"/"List Total" to "Population Hits"/"Population Total." The "EASE score" represents the significance of obtaining "count" genes in the "List Total" for the given KEGG pathway term which has "Population Hits" genes in the background "Population Total."

information on the disease state, GSE1919, GSE12021 (U133A), GSE12021 (U133B), GSE55235 and GSE55457, used tissue from patients with more than 10 years of disease. Patients from the other dataset GSE77298, were at the end stage of the disease. Since osteoclast differentiation is reported in severely inflamed RA synovium, the process is likely to be detected in the synovial tissue used for the datasets. It is noteworthy that our analysis identified the pathway osteoclast differentiation as one of the enriched pathways with a fold enrichment of 3.11 (EASE score of $2.41\ e^{-8}$) in the RA synovium. In addition, our analysis detected the upregulation of two well-known osteoclast markers Cathepsin K and tartrate resistant acid phosphatase 5 (ACP5) in the synovium.

The category highly represented in the list of downregulated pathways was "signal transduction." All the signal transduction pathways enriched in the downregulated gene list are given in Table 5. In contrast to the upregulated pathways, the downregulated

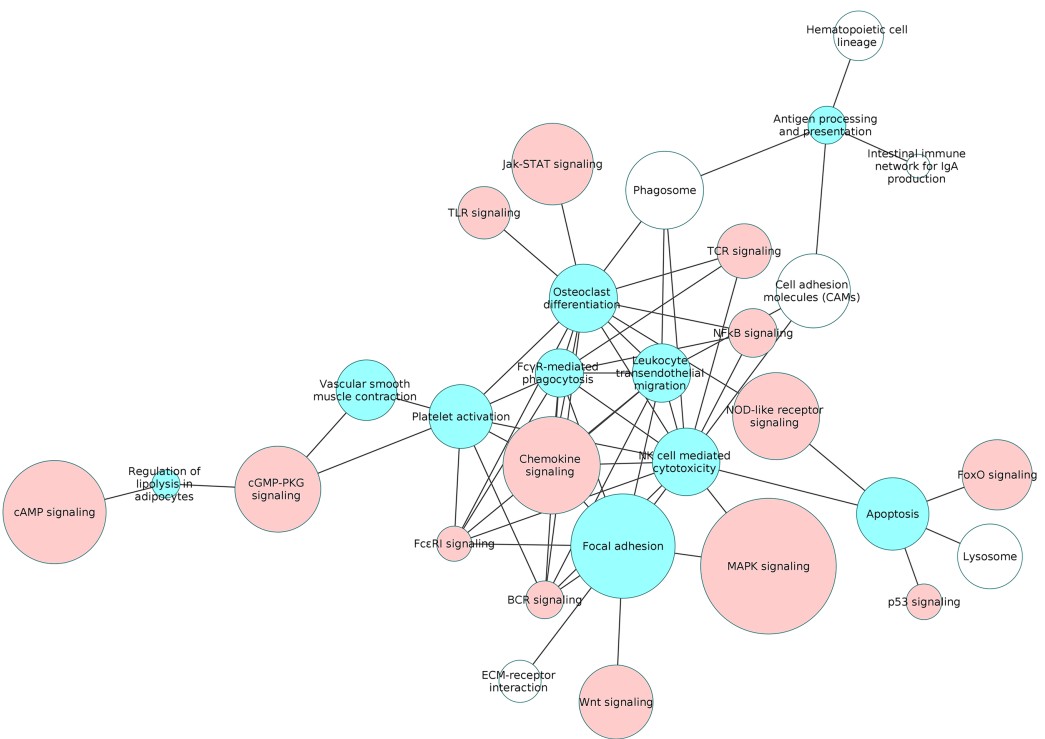

**Figure 6 The interaction network of overlapping pathways.** The blue and red pathway nodes denote the process and the signaling pathways respectively. The white pathway nodes denote the other non-disease pathways. The connection between the pathway nodes represents sharing of at least five upregulated or five downregulated genes by the pair of pathway nodes. The size of the node represents the number of genes present in the pathway. Larger nodes show pathways having more number of genes, and smaller nodes represents pathways with lower number of genes.

pathways included several metabolic pathways such as fatty acid degradation, fatty acid elongation, etc. Some endocrine system pathways like regulation of lipolysis in adipocytes, insulin signaling pathway, which are closely related to metabolic regulation were also listed among the downregulated pathways.

## RA affected signaling pathways interact to orchestrate osteoclast differentiation in the synovium

We categorized the 26 upregulated and 25 downregulated non-disease pathways based on their functional specificity. In this analysis, we identified 12 process pathways and 19 signaling pathways among the differentially regulated pathways. This list of 31 pathways includes the process pathway focal adhesion which was differentially regulated in both directions. While 4 and 10 process and signaling pathways, respectively, were downregulated, 9 process and signaling pathways each were upregulated. The detailed information about the number of upregulated, downregulated and total genes in each of the selected pathway is submitted as Tables S7 and S8.

We examined the overlap of the process pathways with all the non-disease pathways based on the shared number of DEGs. The overlapping pathways are represented as a pathway interaction network in Fig. 6. Several of the signaling pathways share DEGs with

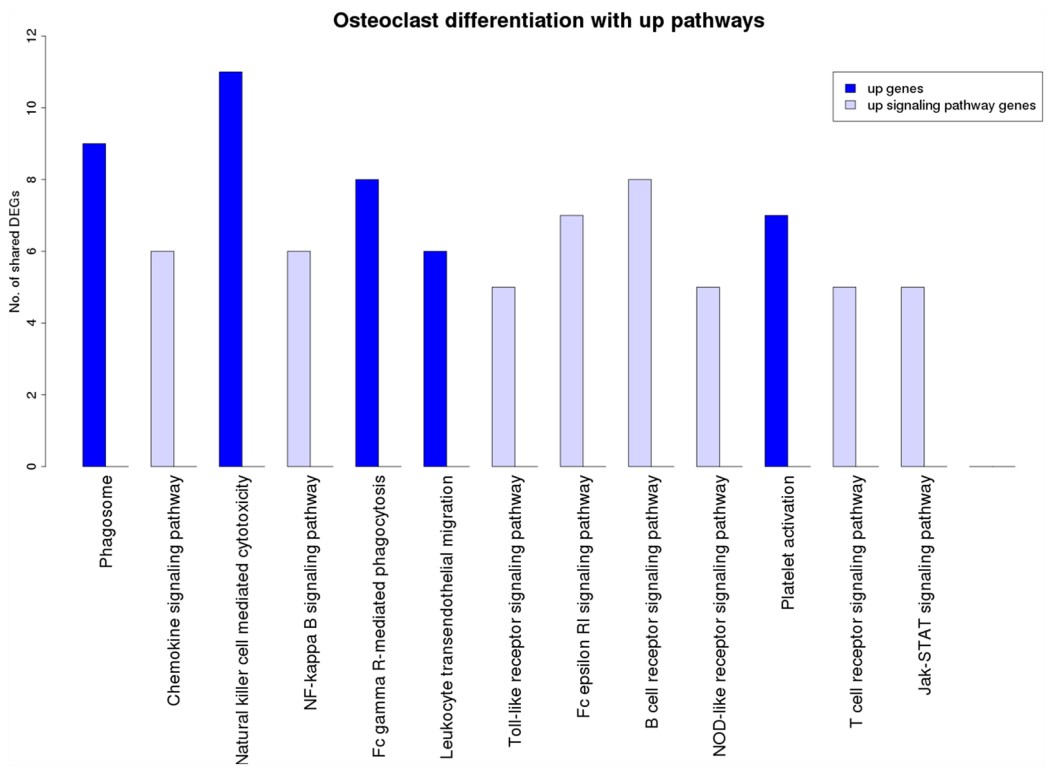

**Figure 7** Overlap analysis of the upregulated process pathway osteoclast differentiation with other enriched non-disease pathways in the RA synovium. The *x*-axis represents the enriched non-disease pathways with which the osteoclast differentiation pathway share DEGs. The *y*-axis denotes the number of DEGs shared between the osteoclast differentiation pathway and each of the enriched non-disease pathways. Upregulated signaling pathway genes are shown in light blue and upregulated non-signaling pathway genes in dark blue. The pathway osteoclast differentiation does not share genes with downregulated pathways.

the process pathways indicating that the process is influenced by these signaling pathways. The details of the DEGs shared by each pathway pair is presented in the Table S9.

Figure 7 is a graphical representation of the number of genes shared between the ODP and other non-disease pathways. Signaling pathways, represented by light blue bars, constitute 8 out of the 13 non-disease pathways that interact with osteoclast differentiation. All the pathways sharing genes with ODPs are upregulated pathways. Similar graphs were created for all upregulated process pathways and are available in the supplementary figures: Figs. S1–S7. Figure 8 shows the overlap analysis for the downregulated signaling pathways.

Among the downregulated process pathways shown in Fig. 8, vascular smooth muscle contraction interacted with platelet activation and cyclic guanosine monophosphate (cGMP)-PKG signaling pathway, via downregulated genes in the RA synovium. Regulation of lipolysis in adipocytes interacted only with down signaling pathways through downregulated genes. Finally, the pathway adherens junction did not overlap with any other pathways.

Our study revealed that the processes of natural killer cell mediated cytotoxicity as well as osteoclast differentiation involved a network of several interacting pathways in the

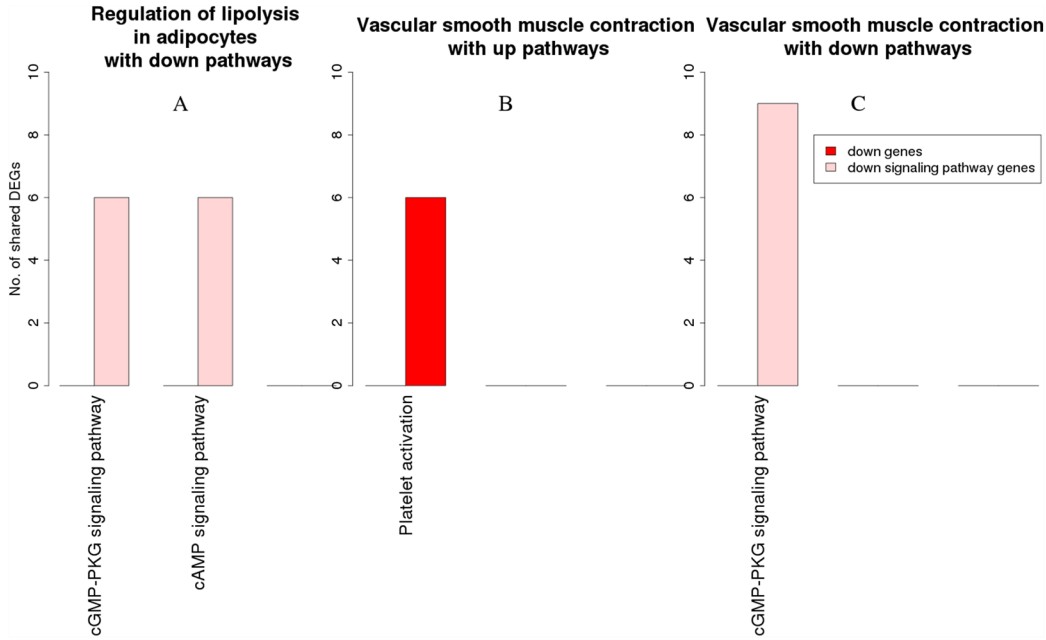

**Figure 8 Overlap analysis of the enriched downregulated process pathways in the RA synovium.**
The *x*-axis represents the enriched non-disease pathways with which the downregulated process pathways share DEGs. The *y*-axis denotes the number of DEGs shared between the downregulated process pathways and each of the enriched non-disease pathways. Downregulated signaling pathway genes are shown in light pink and downregulated non-signaling pathway genes in red. (A) Number of DEGs shared by the pathway "Regulation of lipolysis in adipocytes" with downregulated pathways. (B) Number of DEGs shared by the pathway "Vascular smooth muscle contraction" with upregulated pathways. (C) Number of DEGs shared by the pathway "Vascular smooth muscle contraction" with downregulated pathways.               

RA synovium. However, osteoclast differentiation was influenced by the highest number of signaling pathways. This indicates that the differentiation of osteoclasts in the RA synovium is coordinated by several signaling pathways. In order to understand the collective effect of these signaling pathways on the osteoclast differentiation, we created a detailed PPI network for the process in the RA synovium. Among all the interactions obtained from the STRING database, we used only the experimentally validated ones published in literature, for the creation of the network. In this network, we indicated the differentially regulated genes from the microarray analysis to show the possible ways by which the altered signaling promotes osteoclastogenesis in RA synovium.

## A comprehensive PPI network for the differentiation of osteoclasts in RA synovium

The PPI network, created in our study, had 433 proteins and 1,790 interactions. The network consisted of three connected components. The two smaller connected components were the interactions between CD47 and signal regulatory protein alpha (SIRPA), and IL1A and S100A13.

The protein ubiquitin which has the highest number of interactions in the network was found to interact with 175 network proteins. This is expected, as ubiquitination is a

very common PTM that marks the proteins for proteasomal degradation. In our network, ubiquitination was represented as interaction of a protein with ubiquitin as well as with ubiquitin ligases. We removed ubiquitin from our network since most of the edges of ubiquitin and those of the ubiquitin ligases were redundant. The resulting network had 432 proteins and 1,595 interactions. In this network, in addition to the two small connected components, four proteins, PPP3R1, PPP3CA, PPIA and regulator of calcineurin 1 (RCAN1) were disconnected from the main network and formed a new connected component. The network now had four components: CD47-SIRPA, IL1A-S100A13, PPP3R1-PPP3CA-PPIA-RCAN1, and one large component.

We removed the three smaller connected components from the main network. The large connected component, consisting of 424 protein nodes and 1,589 interactions, was used for further analysis. Henceforth, we refer to this as directed ODP network. The directed ODP network contains 82 core proteins belonging to the KEGG ODP. The portion of the directed ODP network containing the 82 core proteins and their 152 connections is termed as the "**core network**" (directed). The rest of the network consisting of the first-shell interactors and their edges is the "**shell network**" (directed). The core network contains proteins which are directly involved in the osteoclast differentiation. The shell network represents the protein milieu in the RA synovium facilitating the osteoclast differentiation. The complete directed ODP network is provided as a File S1. A second File S2 contains the information about the core and shell proteins of the ODP network.

## The DNA-binding proteins of the directed ODP network

In a PPI network, the terminal responders of the signals are the DNA-binding proteins such as a TFs, coactivator, etc., or the proteins that generate non-protein signaling molecules like secondary messengers. Using the enriched terms in the category GOMF, we classified 82 of the 424 nodes as DNA-binding proteins (Fig. S8). A total of 18 DNA-binding proteins which include STAT, NF-κB and AP1 TF belong to the core network. Along with the differential expression of STAT1, JUN, JUNB and FOSB, the STAT protein STAT2 was upregulated in this study.

In addition to STAT and AP1 proteins, the other DNA binding core protein peroxisome proliferator activated receptor gamma (PPARG) was observed to be downregulated which is in agreement with the results of *Li et al. (2017)*. The attachment of the shell network resulted in the inclusion of nine differentially regulated DNA-binding proteins in the directed ODP network. The possible roles of these proteins in osteoclastogenesis are described in context of their GOBP terms.

## The signaling pathways of the directed ODP network

GO biological process over-representation analysis of the directed ODP network proteins identified several immune signaling terms. These terms included five out of the eight upregulated KEGG signaling pathways which were found to interact with the osteoclast differentiation (Fig. 7). These pathways are: T cell receptor signaling pathway, B cell

receptor signaling pathway, Fc-ε receptor signaling pathway, NF-κB signaling pathway and TLR signaling pathway.

Among these, T cell receptor signaling pathway had the most number of DEGs, with 14 up and one downregulated nodes. The proteins belonging to the T cell receptor signaling term were extracted as a subnetwork (Fig. S9). In this subnetwork, the T cell surface molecules CD3E and CD28 were upregulated whereas CD247 did not show differential regulation. The downstream signaling molecules zeta chain associated protein kinase 70 (ZAP70), LCK, ITK, CSK, LAT, LCP2, FYB, PAG1, PIK3CD, MAPK1, PLCG2 and INPP5D were upregulated. Among them, PIK3CD, MAPK1, LCK, LCP2 and PLCG2 are the core network proteins.

The B cell receptor signaling pathway (Fig. S10) term shared six proteins with the core network. Five of these core proteins were upregulated. In addition, ZAP70, LYN and PRKCB, which are part of the shell network, were found to be upregulated.

The Fc-ε receptor signaling pathway (Fig. S11) showed the MAP Kinase, NF-κB, Rac signaling components. The term showed nine upregulated genes which included the receptor FCER1G.

Six GOBP terms were combined to extract 75 proteins of the NF-κB signaling pathway from the directed ODP network (Fig. S12). This NF-κB subnetwork contained five receptors including one core protein (TNFRSF1A). Out of the 11 DEGs in the subnetwork, TNFSF11 (RANKL) and STAT1 were the core proteins. REL, a component of NF-κB TF dimers was upregulated. The osteoclast differentiation and activation factor, RANKL, an activator of NF-κB pathway was highly upregulated with a log2 fold change of 3.32. It is known that REL participates in the canonical NF-κB signaling (*Shih et al., 2011*). Of the many possible signaling routes leading to REL, we observed DEGs in the following pathways: TCR-PRKCQ-CARD11-BCL10; IL1R1-MyD88-IRAK4; TNFSF11 (RANKL)-TRAF6-IKK; CD27-TRAF2-IKK. We have extracted all these routes and created a subnetwork for activation of REL in the RA synovium (Fig. 9).

We combined nine over-represented GOBP terms to extract the TLR signaling pathway proteins (Fig. S13). The extracted subnetwork featured the signaling from the receptors toll like receptor 3 (TLR3) and toll like receptor 4 (TLR4) to the IκB kinase complex (IKK). Although the pathway is upregulated in the RA synovium, the TLR receptors in the ODP network did not show any differential regulation.

Interestingly, the directed ODP network demonstrated a downregulation of all the DEGs participating in the TGFβ signaling pathway. The TGFβ subnetwork (Fig. S14) showed the presence of the ligands transforming growth factor beta-1 (TGFB1), transforming growth factor beta-2 (TGFB2) and transforming growth factor beta-3 (TGFB3) and the receptors TGF-beta receptor type-1 (TGFBR1), TGF beta receptor type-2 (TGFBR2) and TGF beta receptor type-3 (TGFBR3). Among the downstream SMADs, SMAD3 was downregulated.

## Protein clusters in the directed ODP network

The analysis done using the MCODE application of the Cytoscape tool revealed 19 clusters which may indicate functional protein complexes. The details of these 19 clusters are
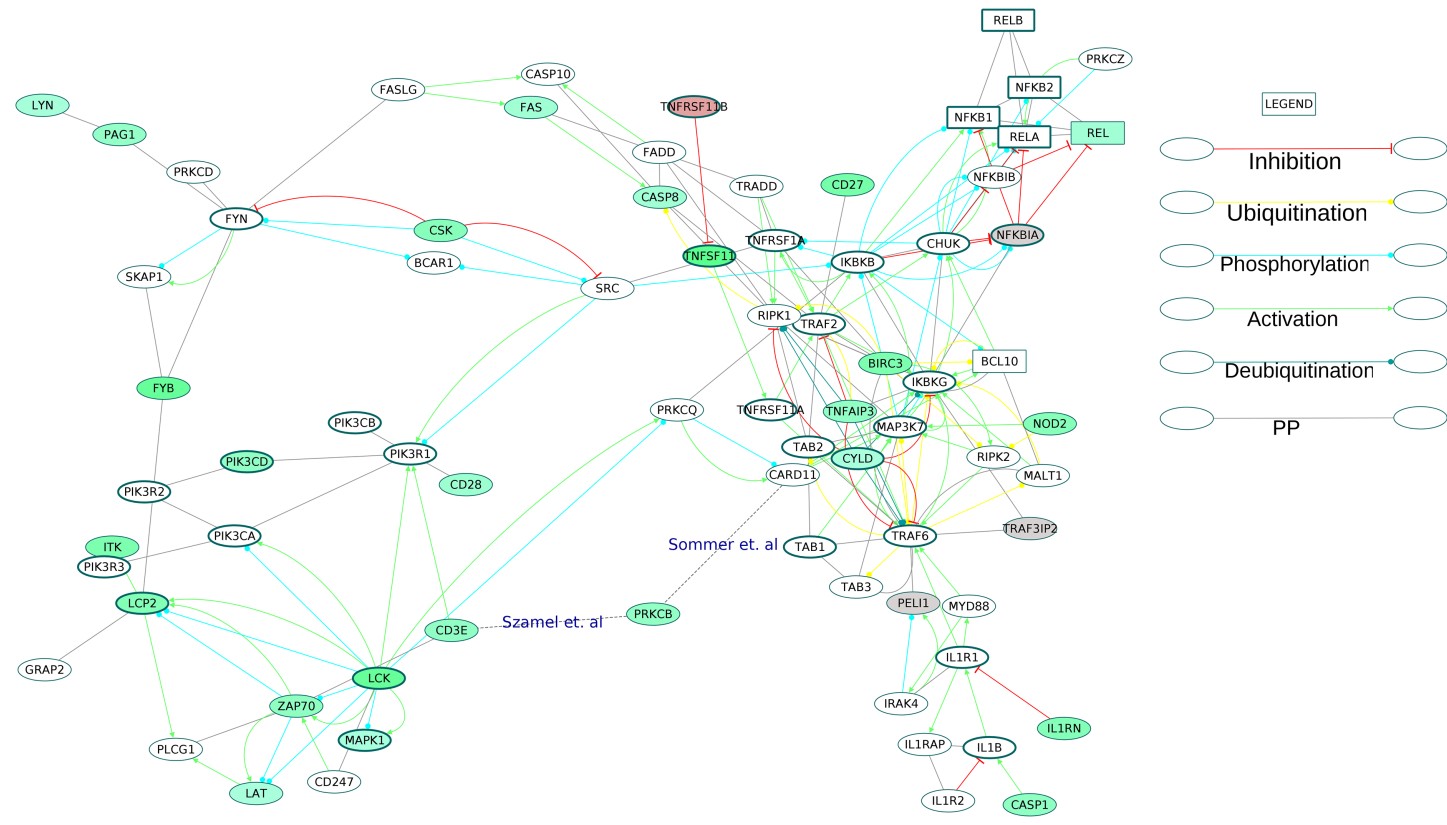

**Figure 9 Activation of NF-κB proteins in the osteoclast differentiation network.** The core proteins are represented by thick borders and the shell proteins by thin borders. Rectangle nodes represent the DNA-binding proteins. The RA drug targets are indicated by red borders. The degree of differential regulation of the nodes is denoted as follows: red to gray–downregulation and green–upregulation. The edges of PRKCB are included from literature.

submitted as Table S10. Among them, 12 clusters included a mixture of core and shell proteins. In three of the clusters (Cluster 2, Cluster 4 and Cluster 10), the core proteins were differentially regulated. Cluster 2 comprised of three core proteins and six shell proteins. The cluster had two DEGs which were DNA-binding proteins STAT1 and PPARG. Cluster 4 was a TGFβ cluster and cluster 10 was an NCF cluster. The TGFβ cluster had two downregulated proteins, of which SMAD3 was a shell protein. The NCF cluster had four core proteins, of which neutrophil cytosol factor 2 (NCF2), neutrophil cytosol factor 4 (NCF4) and cytochrome B-245 alpha chain (CYBA) were upregulated. The upregulated shell protein in the NCF cluster was neutrophil cytosolic factor 1 (NCF1).

## DISCUSSION

This study is aimed at understanding the mechanisms involved in a specific RA-related phenotype. We have used a large number of microarray studies and relaxed inclusion criteria for differential expression across datasets, to obtain relatively large number of DEGs that are likely to be involved in RA. We have combined this gene expression data with pathway analysis and identified various process pathways and several signaling pathways to be affected by RA. In systemic diseases like RA, pathways responsible for a particular phenotype operate in an environment consisting of various other disrupted

pathways. Thus it becomes important to understand the effect of this environment on the pathway immediately responsible for the phenotype. We attempted to achieve this by overlapping the various process pathways with the enriched signaling pathways in the synovium. Interestingly, the process pathway osteoclast differentiation overlapped with several of the enriched signaling pathways. In order to understand the signaling involved in osteoclast differentiation in the RA synovium, for the first time, we created a detailed PPI network responsible for the phenomenon. Each interaction in this network was manually verified from literature enabling the inclusion of directions of the interactions and specific PTMs whenever such information was available. While creating the network using all the possible interactions available in STRING, we found that some proteins in the repository have more number of interactions reported than the others. We acknowledge that this might have led to a bias in the directed ODP network. The network lacks the important non-protein molecules involved in triggering the ectopic differentiation of osteoclasts in the inflamed synovium. In addition, gene expression regulation resulting from activation or repression of TFs was not depicted in the network. Since the RA specific data used in this network was only gene expression data, information on the activation state of specific proteins that are known to be involved in the disease, for example, phosphorylation state of STAT1 was missing. Though the network lacks these information, it is the most comprehensive and informative PPI network till date describing the process of osteoclast differentiation.

## The differentially regulated genes in the RA synovium

In order to identify the DEGs in the RA synovium, seven microarray datasets generated by five different studies were used. Among the seven datasets, the RA patients belonging to the datasets GSE1919, GSE12021 (U133A), GSE12021 (U133B), GS55235 and GSE55457 had similar high values for the inflammatory markers, ESR and CRP. Additionally, the tissue used in GSE77298 were described as end stage RA synovial biopsies. Therefore, we surmise that the RA tissues were highly inflamed. However, we observed that few genes were differentially expressed across most of the datasets (Fig. 2). Since the level of inflammation in RA tissues were comparable, we attribute this lack of concordance between the datasets to the heterogeneity of the disease.

## The enriched pathways in the RA synovium

Kyoto Encyclopedia of Genes and Genomes pathway enrichment analysis of the upregulated genes (common-up) resulted in 26 upregulated disease pathways. As expected, RA was one of these disease pathways. Staphylococcus aureus infection and Tuberculosis, the two upregulated infectious disease pathways in the results of the pathway enrichment analysis are known to be associated with RA (Sams et al., 2015; Jeong et al., 2017). The upregulated infectious disease pathways share several genes with immune system pathways. Differential regulation of the immune pathways is expected in RA since it is an immune disorder. Therefore, the upregulation of the immune genes explains the enrichment of the infectious disease pathways in the up pathway list.
Several pathways belonging to the category "immune system" were enriched in the common-up genes. The enrichment of these immune pathways is likely as infiltration of activated immune cells has been observed in the RA synovium (*McInnes & Schett, 2011*). In addition, the resident cells of the inflamed pannus exhibit activation of several immune signaling pathways (*McInnes & Schett, 2007*). Among the pathways from Table 4, the Fc-ε RI signaling pathway shows an upregulation of the immunoglobulin E (IgE) receptor Fc-ε RI. Elevated presence of IgE and activated mast cells have been detected in the RA synovium (*Gruber, Ballan & Gorevic, 1988*; *Tetlow & Woolley, 1995*). It has been demonstrated by in vitro experiments that the RA synovial mast cells express Fc-ε RI and can be activated via the Fc-ε RI signaling pathway (*Lee et al., 2013*). However, the contribution of the pathway to the pathological alteration of synovial tissue function needs to be addressed in future studies.

Among the enriched downregulated pathways, seven belong to the category "signal transduction" (Table 5). The AMPK signaling pathway which is known for its anti-inflammatory effect shows downregulation in our analysis (*Speirs et al., 2018*). The downregulation of the FOXO signaling pathway is also observed in this study. This reflects the results of the earlier studies which have shown downregulation and inactivation of the FOXO TFs in the RA affected synovium (*Ludikhuize et al., 2007*; *Grabiec et al., 2015*). Additionally, the FOXO proteins are inhibited by NF-κB signaling, and activated by AMPK as shown in the FOXO signaling pathway listed in KEGG pathway database. The pathway enrichment results from our analysis show upregulation in NF-κB signaling and downregulation in AMPK signaling, which explains the downregulation of FOXO signaling. Further, the cGMP-PKG signaling is known to be essential for the vascular smooth muscle response to the inflammatory cytokines (*Browner, Sellak & Lincoln, 2004*). The downregulation of genes in this pathway needs to be studied in detail to analyze the effects of RA synovial inflammation on the blood vessels in the affected tissue. The cAMP signaling pathway is known to facilitate regulatory T cell function and T cell anergy (*Raker, Becker & Steinbrink, 2016*). The downregulation in this pathway might indicate the pro-inflammatory nature of the infiltrating T cells in the synovium. It is difficult to explain the importance of downregulation in the Wnt signaling pathway and the MAPK signaling pathway in the RA synovium. Wnt signaling, which is required for repair of bone erosions, is suppressed in mouse models of inflammatory arthritis (*Lories, Corr & Lane, 2013*). However, Wnt signaling has also been linked to pro-inflammatory cytokine production in the affected synovium (*Miao et al., 2013*). In our study, the KEGG Wnt signaling pathway is enriched in downregulated genes. Among the downregulated genes, two are Wnt receptors FZD4 and LRP6, whereas three are the Wnt antagonists secreted frizzled related protein 1, secreted frizzled related protein 2 and SFRP. Thus no conclusions can be made about the role of Wnt signaling pathway from its presence in the list of downregulated pathways. Similarly, no clear picture can be drawn about the role of MAPK signaling pathway in the RA synovium. This is because the KEGG MAPK signaling pathway is very large, with 252 genes and several genes are grouped together under terms like receptor tyrosine kinase and GF, making the pathway extremely general.

## Construction of the directed ODP network

Three small connected components were removed while creating the directed ODP network. One of the components contained the proteins PPP3R1 and PPP3CA which are the subunits of calcineurin, an important regulator of osteoclast differentiation. The immunosuppressant drug cyclosporin used in the treatment of RA acts as an inhibitor of calcineurin by forming a ternary complex with PPIA and calcineurin (*Wang & Heitman, 2005*). The remaining protein in the connected component, RCAN1 is an inhibitor of calcineurin. Both in the KEGG pathway database and in literature, these proteins are described as participating in the RANKL signaling pathway which regulates the differentiation of osteoclasts. Calcineurin participates in the RANKL signaling pathway downstream of the non-protein components inositol triphosphate and calcium ions. Since our network is based on PPIs, it failed to capture the connection of calcineurin and its adjacent proteins to the large connected component, in spite of showing the connections of other RANKL pathway proteins.

## Analysis of the directed ODP network

The directed ODP network was analyzed to examine three main aspects. Firstly, the proteins binding to DNA were explored because they are involved in regulating gene expression. Secondly, the proteins involved in signal transduction were studied for their role in facilitating the process of osteoclast differentiation in the RA affected synovium. Finally, the clusters of highly interconnected proteins were identified in the network. Several proteins function as part of protein complexes. The protein clusters we identified may represent the protein complexes. We also examined the differential expression of the proteins in the clusters. Protein clusters with differentially regulated proteins may represent complexes actively involved in the osteoclastogenesis in RA synovium.

## DNA binding proteins in the directed ODP network

Among the DNA-binding proteins in the directed ODP network, the downregulation of the AP1 proteins JUN and JUNB is in contrast to the earlier studies which reported their upregulation in RA synovium (*Kinne et al., 1995*). As shown in Table 3, the AP1 downregulation was observed in five of the seven datasets, with the three proteins FOSB, JUN and JUNB being downregulated in three out of the seven datasets. The presence of consistent downregulation in the datasets from different studies shows that the downregulation of these proteins is not due to dataset specific factors. As mentioned earlier, four of the five datasets showing downregulation had patients with similar clinical characteristics. This may suggest a connection between the stage of the disease and the downregulation of the AP1 proteins. Further studies are required to explain the downregulation of these AP1 proteins in the synovium of a subset of RA patients.

According to previous reports, another AP1 protein FOS is upregulated in the RA synovium (*Dooley et al., 1996*). However, our analysis does not show any differential regulation of FOS. FOS is an indispensable TF for osteoclast differentiation (*Grigoriadis et al., 1994*). In the ODP network, the upregulation of MAPK1 and RPS6KA1 presents a

possible mechanism for FOS activation. In addition, our network reveals the activation of FOS by the upregulated TF STAT1. Along with STAT1, the TF STAT2, the kinase JAK2 and the receptor IFNAR2 were upregulated in the IFN pathway of the ODP network. TNFSF11 (RANKL) induces expression of interferon beta (IFNβ) which serves as a feedback inhibitor of osteoclastogenesis via STAT1 (*Xiong et al., 2016*). The upregulation of the IFNβ receptor, IFNAR2 and the TF STAT1 may indicate that the feedback inhibition is functional in the RA synovium.

## Signaling pathways in the directed ODP network

The examination of the enriched GOBP terms of the network showed an involvement of T cell receptor signaling pathway, B cell receptor signaling pathway, Fc-ε receptor signaling pathway and NF-κB signaling pathway in the differentiation of osteoclasts in the RA synovium. Figure 9 depicts the signaling routes that lead to activation of REL in the RA synovium. The figure shows activation of REL by PKC. The network shows T cell receptor mediated activation of PRKCQ and its subsequent activation of CARD11. Based on earlier studies (*Szamel, Bartels & Resch, 1993*; *Sommer et al., 2005*), we speculate that another isoform of PKC, PRKCB is also involved in the activation of CARD11 in the affected synovium. It is known that the activated CARD11 via the formation of a trimeric complex with BCL10 and MALT1 activates the IKK complex (*Turvey et al., 2014*). The IKK complex in turn activates NF-κB proteins including REL by removing their inhibition by Nf-κb inhibitor alpha (NFKBIA) and Nf-κB inhibitor beta (NFKBIB). Figure 9 shows that the ODP network has captured the interactions of these proteins. In this pathway, the upregulation of PRKCB and REL and the downregulation of NFKBIA indicate activation of REL via this pathway in the RA affected synovium. It is also known that NF-κB proteins are activators of RANKL (TNFSF11) gene expression in activated T cells (*Fionda et al., 2007*). Our network, through the presence of DEGs in the RA synovium, shows how TCR signaling aids in activation of REL which leads to TNFSF11 expression and osteoclastogenesis. In addition, the upregulation of TNFSF11 and downregulation of its competitive inhibitor TNFRSF11B marks another route to the activation of REL via TRAF6. PPARG, a known inhibitor of TNFSF11-mediated osteoclastogenesis, was downregulated in the ODP network. This result agrees with the findings of *Li et al. (2017)*. The upregulation of CD27 receptor is in accordance with the reports of high levels of CD27 in the synovial tissue of RA patients (*Tak et al., 1996*). As CD27 also activates TRAF6, we speculate that the upregulation of CD27 contributes to the activation of REL via this route in the RA synovium. Among the DEGs involved in the REL activation via IL1 pathway, caspase 1 (CASP1) was upregulated whereas the inhibitor PELI1 was downregulated. This is balanced against the upregulation of IL1RN, a competitive inhibitor of IL1R1. It is known that both the cytokines, IL1 and TNF, activate NF-κB through TRAF6. However, we did not observe differential regulation of the TNF pathway proteins leading to the activation of NF-κB. On the other hand, the apoptosis related proteins such as FAS and CASP8 which are downstream to TNF receptor, were upregulated. Earlier studies have established that the death signaling pathways are antagonized by the activity of BIRC2, BIRC3 and X-linked inhibitor of apoptosis (XIAP)

(*Vasudevan & Ryoo, 2015*). The upregulation of BIRC3 points to suppression of TNF mediated apoptosis and the activation of NF-κB via TNF receptor signaling in the RA synovium. This reflects the possibility that TNF signaling results in both CASP8 mediated apoptosis and BIRC3 mediated NF-κB activation in different parts of the RA synovium. The network also captures the activation of IKK by the upregulated NOD2 through MAP3K7. Studies have reported that RA synovial cells express high levels of NOD2 (*Franca et al., 2016*). We hypothesize that NOD-dependent activation of NF-κB also contributes to the osteoclastogenesis in RA synovium. All the routes that lead to activation of NF-κB points to the canonical signaling. PEL1 is known to be a negative regulator of REL (*Chang et al., 2011*). REL, which functions only in the canonical NF-κB pathway, is the only NF-κB protein showing differential regulation in this analysis (*Shih et al., 2011*). The upregulation of REL, BIRC3 and PRKCB and the downregulation of PELI1 also support the predominance of canonical NF-κB signaling in RA synovium osteoclastogenesis (*Lutzny et al., 2013*; *Varfolomeev et al., 2007*).

The GOBP analysis revealed enrichment of TGFβ signaling pathway terms. The role of TGFβ signaling pathway in osteoclastogenesis is not captured in the KEGG ODP as it shows only the receptor-ligand interaction. Our network connects the TGFβ receptors to downstream DNA-binding proteins, of which Forkhead box proteins, SMAD3 and ATF3 were downregulated. The consistent downregulation of all the DEGs in the TGFβ pathway implies their negative regulatory role in the osteoclastogenesis. Although the published studies support a mixed role of TGFβ pathway in the RA synovium, *Karst et al. (2004)* showed that high concentration of TGFβ is involved in the inhibition of osteoclastogenesis. Therefore, we hypothesize that the downregulation of the TGFβ pathway produces a favorable environment for osteoclastogenesis in the RA synovium.

## A proposed model of enhanced ROS production mediating osteoclastogenesis in RA synovium

It is known that the NADPH oxidase 2 (Nox2) complex generates ROS which act as secondary messengers during osteoclast differentiation (*Kang & Kim, 2016*). ROS are also known to cause activation of canonical NF-κB pathway (*Gloire, Legrand-Poels & Piette, 2006*). In the ODP network, the upregulation of NCF1, NCF2, NCF4 and CYBA, the four components of Nox2 complex, may indicate the osteoclast differentiation as well as oxidative burst by phagocytic cells in the RA synovium (*Rosen et al., 1995*). The NCF cluster, selected by MCODE analysis, demonstrates the activation of the core proteins NCF2, NCF4, CYBA and RAC1 by the shell proteins NCF1, PRKCZ and PARD6G. It is known that PRKCZ is activated by T cell receptor (*Bertrand et al., 2010*). In our analysis, it was observed that multiple T cell receptor signaling molecules were upregulated. We hypothesize that the TCR signaling via PRKCZ activates NCF complex which may subsequently generate ROS in the RA synovium. It is known that ROS can diffuse across cell membranes to take part in intracellular signaling (*Fisher, 2009*). We believe that the activation of PRKCZ leading to the generation of ROS is one of the routes facilitating osteoclast differentiation in RA synovium. Our network also illustrated an upregulation

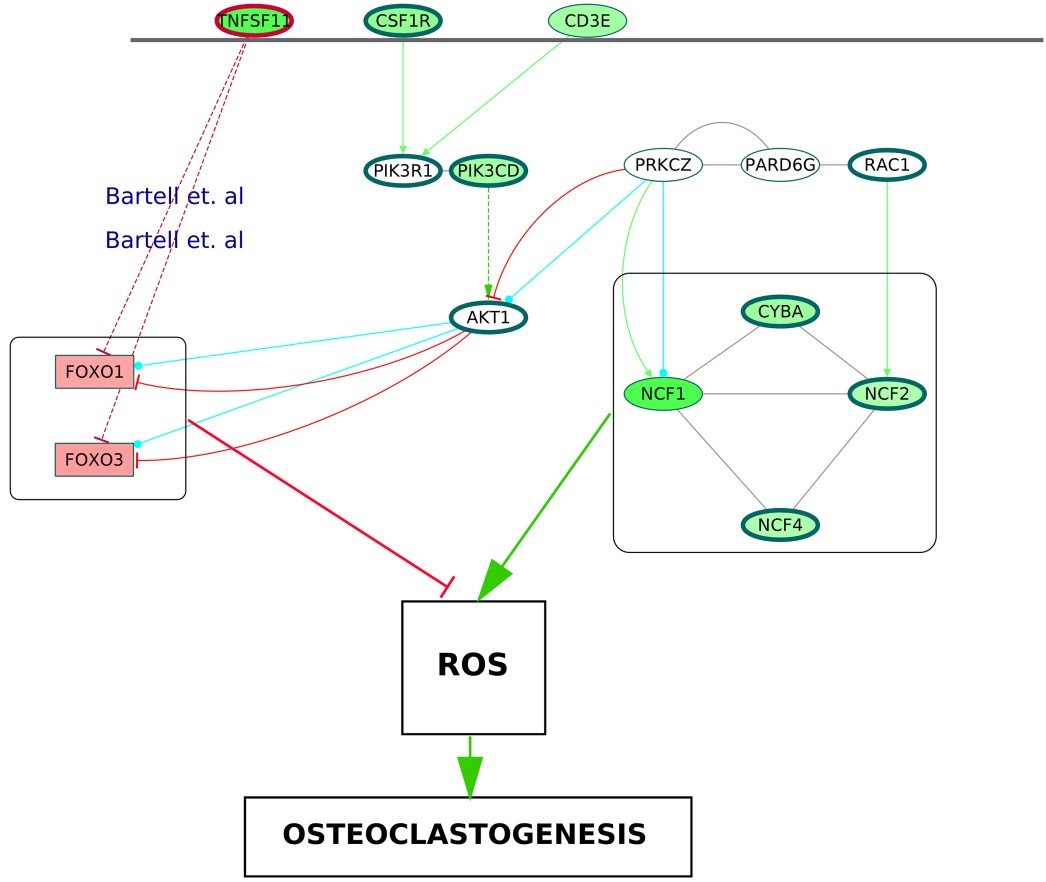

**Figure 10 A schematic of enhanced ROS production mediating differentiation of osteoclasts in the RA synovium.** The degree of differential regulation of the nodes in the RA synovium is denoted as follows: red–downregulation and green–upregulation. The cytokine (TNFSF11) and the membrane receptors (CSF1R and CD3E) are above the horizontal line. Genes corresponding to the Nox2 complex and the Forkhead Box proteins are within rectangular boxes. The colors of the edges denote the following: green–activation, red–inhibition, blue–phosphorylation and black–PP. The dotted edges represent the interactions which involve other intermediate molecules. The inhibition of Forkhead box proteins by TNFSF11 are included from literature.           

of the catalytic subunit of PI3K, PIK3CD which is another downstream molecule of T cell receptor signaling. This is supported by the findings of *Bartok et al. (2012)*. PI3K-AKT pathway results in inhibition of the Forkhead box proteins, FOXO1 and FOXO3 (*Patel & Mohan, 2005*). It was shown that FOXO upregulates antioxidant enzymes that inhibit osteoclastogenesis (*Bartell et al., 2014*). In the ODP network, the downregulation of FOXO1 and FOXO3 is a possible indication of the presence of ROS mediated osteoclastogenesis in RA synovium. *Bartell et al. (2014)*, experimentally proved the role of TNFSF11 in the downregulation of the Forkhead box proteins. In accordance with this, our analysis showed an upregulation of TNFSF11 in the ODP network. Finally, our analysis reported the upregulation of the cytokine receptor colony stimulating factor 1 receptor (CSF1R) which is also required for osteoclastogenesis via PI3K-AKT pathway. The proposed model consisting of all the signaling routes promoting osteoclastogenesis via generation of ROS in the RA synovium is summarized in Fig. 10.

## CONCLUSION

In this study, we have created a PPI network for osteoclast differentiation in the RA synovium for the first time using gene expression under RA conditions from microarray experiments, pathway enrichment analysis and PPI data. This network captures all the signaling routes that lead to osteoclastogenesis in the synovium and depicts the roles of T cell receptor signaling, canonical NF-κB pathway and ROS generation.

## ABBREVIATIONS

| | |
|---|---|
| ACP5 | Acid Phosphatase 5, Tartarate Resistant |
| AKT | Akt Serine/Threonine Kinase 1 (Protein Kinase B) |
| AMPK | 5′ Adenosine Monophosphate-Activated Protein Kinase |
| AP1 | Activator Protein 1 |
| BCL10 | B Cell Cll/Lymphoma 10 |
| BIRC2 | Baculoviral Iap Repeat-Containing Protein 2 (Cellular Inhibitor Of Apoptosis 1) |
| BIRC3 | Baculoviral Iap Repeat-Containing Protein 3 (Cellular Inhibitor Of Apoptosis 2) |
| CARD11 | Caspase Recruitment Domain Family Member 11 |
| CASP1 | Caspase 1 |
| CD247 | T-Cell Surface Glycoprotein CD3 Zeta Chain |
| CD27 | T-Cell Activation Antigen CD27 |
| CD28 | T-Cell-Specific Surface Glycoprotein CD28 |
| CD3E | T-Cell Surface Glycoprotein CD3 Epsilon Chain |
| CD47 | Leukocyte Surface Antigen CD47 (Integrin Associated Protein) |
| cGMP | Cyclic Guanosine Monophosphate |
| CRP | C-Reactive Protein |
| CSF1R | Colony Stimulating Factor 1 Receptor |
| CSK | C-Terminal Src Kinase |
| CTSK | Cathepsin K |
| CYBA | Cytochrome B-245 Alpha Chain |
| DAVID | Database for Annotation, Visualization and Integrated Discovery |
| DEG | Differentially expressed gene |
| EASE | Expression Analysis Systematic Explorer |
| ECM | Extra cellular matrix |
| ESR | Erythrocyte Sedimentation Rate |
| FAS | Fas cell Surface Death Receptor (CD95) |
| FCER1G | High Affinity Immunoglobulin Epsilon Receptor Subunit Gamma 3 |
| FOS | Fos proto-oncogene |
| FOSB | FosB Proto-Oncogene, AP-1 Transcription Factor Subunit |
| FOXO1 | Forkhead Box Protein O1 |
| FOXO3 | Forkhead Box Protein O3 |

| | |
|---|---|
| **FYB** | Fyn Binding Protein |
| **FZD4** | Frizzled Class Receptor 4 |
| **GEO** | Gene Expression Omnibus |
| **GF** | Growth Factor |
| **GO** | Gene Ontology |
| **GOBP** | Gene Ontology Biological Process |
| **GOMF** | Gene Ontology Molecular Function |
| **GRN** | Gene Regulatory Network |
| **ID** | Identifier |
| **IFNAR2** | Interferon Alpha and Beta Receptor Subunit 2 |
| **IFNβ** | Interferon Beta |
| **IgE** | Immunoglobulin E |
| **IGKC** | Immunoglobulin Kappa Constant Region |
| **IKK** | Inhibitor of Nuclear Factor Kappa B Kinase |
| **IL1A** | Interleukin 1 Alpha (hematopoietin 1) |
| **IL1R1** | Interleukin 1 Receptor Type 1 |
| **IL1RN** | Interleukin 1 Receptor Antagonist |
| **IL7R** | Interleukin 7 Receptor |
| **INPP5D** | Inositol Polyphosphate-5-Phosphatase D |
| **IκB** | Nf-Kappa-B Inhibitor |
| **IRAK4** | Interleukin 1 Receptor Associated Kinase 4 |
| **ITK** | Il2 Inducible T Cell Kinase |
| **JAK2** | Janus Kinase 2 |
| **JUN** | Jun proto-Oncogene, AP-1 Transcription Factor Subunit |
| **JUNB** | JunB Proto-Oncogene, AP-1 Transcription Factor Subunit |
| **KEGG** | Kyoto Encyclopedia of Genes and Genomes |
| **LAT** | Linker for Activation of T Cells |
| **LCK** | Lymphocyte-Specific Protein Tyrosine Kinase |
| **LCP2** | Lymphocyte Cytosolic Protein 2 (Sh2 Domain Containing Leukocyte Protein of 76kda) |
| **LRP6** | Ldl Receptor Related Protein 6 |
| **LYN** | Lck/Yes-Related Novel Protein Tyrosine Kinase |
| **MAP3K7** | Mitogen-Activated Protein Kinase Kinase Kinase 7 (TGF-Beta Activated Kinase 1) |
| **MAPK1** | Mitogen-Activated Protein Kinase 1 |
| **MAS5** | Microarray Suite 5.0 |
| **MYD88** | Myeloid Differentiation Primary Response 88 |
| **NCF** | Neutrophil Cytosol Factor |
| **NCF1** | Neutrophil Cytosolic Factor 1 |
| **NCF2** | Neutrophil Cytosol Factor 2 |
| **NCF4** | Neutrophil Cytosol Factor 4 |

| | |
|---|---|
| **NF-κB** | Nuclear Factor Kappa-Light-Chain-Enhancer Of Activated B Cells |
| **NFKBIA** | Nf-κb Inhibitor Alpha |
| **NFKBIB** | Nf-κb Inhibitor Beta |
| **NOD2** | Nucleotide Binding Oligomerization Domain Containing Protein 2 |
| **Nox2** | NADPH Oxidase 2 |
| **ODP** | Osteoclast Differentiation Pathway |
| **PAG1** | Phosphoprotein Associated with Glycosphingolipid-enriched microdomains 1 |
| **PARD6G** | Par-6 Family Cell Polarity Regulator Gamma |
| **PELI1** | Pellino E3 Ubiquitin Protein Ligase 1 |
| **PI3K** | Phosphatidylinositol-4,5-Bisphosphate 3-Kinase |
| **PIK3CD** | Phosphatidylinositol-4,5-Bisphosphate 3-Kinase Catalytic Subunit Delta |
| **PIK3CD** | Phosphatidylinositol-4,5-Bisphosphate 3-Kinase Catalytic Subunit Delta |
| **PKG** | cGMP Dependent Protein Kinase (protein Kinase G) |
| **PLCG2** | Phospholipase C Gamma 2 |
| **PPARG** | Peroxisome Proliferator Activated Receptor Gamma |
| **PPI** | Protein–Protein Interaction |
| **PPIA** | Peptidylprolyl Isomerase A (cyclophilin A) |
| **PPP3CA** | Protein Phosphatase 3 Catalytic Subunit Alpha (calcineurin A Alpha) |
| **PPP3R1** | Protein Phosphatase 3 Regulatory Subunit B, Alpha (calcineurin Subunit B Type 1) |
| **PRKCB** | Protein Kinase C Beta |
| **PRKCQ** | Protein Kinase C Theta |
| **PRKCZ** | Protein Kinase C Zeta |
| **PTM** | Post Translational Modification |
| **RA** | Rheumatoid Arthritis |
| **RAC1** | Rac Family Small GTPase 1 |
| **RANKL** | Receptor Activator of Nuclear Factor Kappa B Ligand |
| **RCAN1** | Regulator of Calcineurin 1 |
| **REL** | Rel proto-Oncogene, Nf-κb Subunit |
| **RMA** | Robust Multiarray Average |
| **ROS** | Reactive Oxygen Species |
| **RPS6KA1** | Ribosomal Protein S6 Kinase A1 |
| **RTK** | Receptor Tyrosine Kinase |
| **S100A13** | S100 Calcium-Binding Protein A13 |
| **SFRP1** | Secreted Frizzled Related Protein 1 |
| **SFRP2** | Secreted Frizzled Related Protein 2 |
| **SFRP4** | Secreted Frizzled Related Protein 4 |
| **SIRPA** | Signal Regulatory Protein Alpha |
| **SMAD3** | Mothers against Decapentaplegic Homolog 3 |
| **STAT1** | Signal Transducer and Activator of Transcription 1 |

| | |
|---|---|
| **STAT2** | Signal Transducer and Activator of Transcription 2 |
| **STRING** | Search Tool for the Retrieval of Interacting Genes/Proteins |
| **TCR** | T Cell Receptor |
| **TF** | Transcription Factor |
| **TGFB1** | Transforming Growth Factor Beta-1 |
| **TGFB2** | Transforming Growth Factor Beta-2 |
| **TGFB3** | Transforming Growth Factor Beta-3 |
| **TGFBR1** | TGF-Beta Receptor Type-1 |
| **TGFBR2** | TGF-Beta Receptor Type-2 |
| **TGFBR3** | TGF-Beta Receptor Type-3 |
| **TGFβ** | Transforming Growth Factor Beta |
| **TLR** | Toll Like Receptor |
| **TLR3** | Toll Like Receptor 3 |
| **TLR4** | Toll Like Receptor 4 |
| **TNFRSF11B** | TNF Receptor Superfamily Member 11b (Osteoprotegerin) |
| **TNFRSF1A** | Tumor Necrosis Factor Receptor 1 |
| **TNFSF11** | Tumor Necrosis Factor Superfamily Member 11 (RANKL) |
| **TRAF2** | TNF Receptor Associated Factor 2 |
| **TRAF6** | TNF Receptor Associated Factor 6 |
| **TRAP** | Tartrate Resistant (Tartrate-Resistant Acid Phosphatase) |
| **UBC** | Ubiquitin C |
| **XIAP** | X-Linked Inhibitor of Apoptosis |
| **ZAP70** | Zeta Chain Associated Protein Kinase 70 |

### Funding

The authors were financially supported by the following funding agencies: Shilpa Harshan by IBAB; Poulami Dey by the Indian Council of Medical Research (ICMR), India, Grant No.—3/1/3 JRF-2013/HRD-106 (20835); Srivatsan Raghunathan by the Department of Biotechnology, Government of India, Grant No.—BTPR 12422/MED/31/287/2014. The project was also supported by the Department of IT, BT and S&T of the Government of Karnataka. The funders had no role in study design, data collection and analysis, decision to publish, or preparation of the manuscript.

### Grant Disclosures

The following grant information was disclosed by the authors:
IBAB.
Indian Council of Medical Research (ICMR), India: 3/1/3 JRF-2013/HRD-106 (20835).
Department of Biotechnology, Government of India: BTPR 12422/MED/31/287/2014.
Department of IT, BT and S&T of the Government of Karnataka.

## Competing Interests

The authors declare that they have no competing interests.

## Author Contributions

- Shilpa Harshan conceived and designed the experiments, performed the experiments, analyzed the data, contributed reagents/materials/analysis tools, prepared figures and/or tables, authored or reviewed drafts of the paper, approved the final draft.
- Poulami Dey conceived and designed the experiments, performed the experiments, analyzed the data, contributed reagents/materials/analysis tools, prepared figures and/or tables, authored or reviewed drafts of the paper, approved the final draft.
- Srivatsan Ragunathan conceived and designed the experiments, analyzed the data, contributed reagents/materials/analysis tools, authored or reviewed drafts of the paper, approved the final draft.

## Data Availability

All the data generated and used for the analysis are included in the article and in the Supplementary Files.

## Supplemental Information

Supplemental information for this article can be found online at http://dx.doi.org/10.7717/peerj.5743#supplemental-information.

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
