# Peer review of "Effects of rheumatoid arthritis associated transcriptional changes on osteoclast differentiation network in the synovium"

_PeerJ, doi:10.7717/peerj.5743_

## Round 0.1 · original submission · Major Revisions

Your manuscript is interesting but there are many concerns in its current form. A thorough revision based on reviewer's suggestions is required.

·

Basic reporting

This is an interesting manuscript which gives ue a lot information about the effect of inflammatory synovium to osteoclast formation. The manuscript is writen in quite professional English, gives sufficient introduction and background to support the work, as well as references. As for results, the authors provide 16 figs and tables, respectively.

Experimental design

The authers got the original data on gene expression, pathways and protein interactions related to rheumatoid arthritis from literature and databases namely Gene Expression Omnibus, KEGG pathway and STRING, and created a network for the differentiation of osteoclasts.

Validity of the findings

NF-κB pathway is well known to related with RA, no matter synovium or osteoclast, as well as TLRs, TGF-β, so this paper is lack of innovativeness.

Additional comments

The role of synovium on osteoclast is still unclear, we don't know inflammatory synovium induce osteoclast differentiation or osteoclast activation leads to synovium inflammation. This paper analysed the related signaling pathway in inflammatory synovium which would have a potential effect on osteoclast activation, interesting bioinformatics work and gave some information to investigate the relationship between synovium and osteoclast. But the problem is that the findings in this manuscript was something well investigated to be key proteins and pathways in osteoclast differentiation, maybe inflammation synovium will induce osteoclast, but needs more works to comfirm.

Reviewer 2 ·

Basic reporting

No comments. Please see general comments to author.

Experimental design

No comments. Please see general comments to author.

Validity of the findings

No comments. Please see general comments to author.

Additional comments

Manuscript 27887v1
Effects of rheumatoid arthritis associated transcriptional changes on osteoclast differentiation network in the synovium

In this manuscript by S. Harshan et al, the authors perform a bioinformatics investigation of gene expression in Rheumatoid Arthritis synovium based on publicly available data. The aim is to further understand the complexity of pathology in inflamed joints and how this could be targeted by therapy. The strategy is interesting, but the manuscript needs to be improved in terms of presentation in order to convincingly convey the conclusions. The study presented may be interesting for researchers in the field of rheumatology, but considering that few of them are bioinformaticians, the paper, in addition, needs clarification and more background information.

(a) The manuscript should have an abbreviation list.
(b) The authors should provide a figure with a flow chart of the work; the rational for using certain databases and algorithms, selection criteria (inclusion and exclusion) in each step, and better explain the interpretation of the networks created. For instance, why was the category “metabolism” not considered in the KEGG database? Why did the authors decide to identify proteins binding to DNA (line 161-162)? etc.
(c) Although this is not a review article, in order to make it easier for readers not completely familiar with the bioinformatic tools, the authors may add a “toolbox” explaining the different databases and their features. It is briefly explained in the text, but the manuscript would benefit from having this information also in a condensed way.
(d) The Supplementary tables need self-explanatory headings.
(e) The Supplementary figures should have legends explaining what is shown in the figures. In general, all Figure legends should be improved in order to better explain the figures.
(f) line 42-43: “Studies have used pathway analysis to identify affected pathways from lists of DEGs” This statement should be followed by references.

Results:
(g) line 185: only STAT1, IL7R, IGKC were upregulated in all seven datasets. This should be elaborated on in the Discussion part. In general, the origin of the data sets should be more critically considered and discussed. It says in line 186 “indicating the heterogeneity of the disease”. Is it rather the heterogeneity of the data sets? It is important to know more about the data sets in order to appreciate, for example, the downregulation of the AP-1 proteins in some of the datasets, but not all of them.
(h) in line 194 is written “two pathways were new” Is new compared to what? In what context? Many parts in the text leaves the reader with questions like that and the authors have to make a thorough editing in order to make the text clearer, and hence, easier for the reader to appreciate the interesting and important parts.
(i) line 209: the authors should explain what is meant with “affected” pathways?
(j) Table 3: How are the listed pathways ordered?
(k) lines 227-230: if known, the authors should comment on whether all patients with more than 10 years of disease still had active synovial inflammation.
(l) Table 5 could be taken out and the information written in the text.
(m) Figure 6: the content and text are very small. In addition, as with all figure legends in this manuscript, the text needs to be more explanatory. It is very difficult to overview what is shown in the figure. Could the data be shown in a different way?
(n) line 256: the expression “up process pathways” needs to be expressed in a way so that it becomes immediately clear for the reader what is meant. This type of expression can be found in other parts of the manuscript as well and should be edited.
(o) what is written in lines 256-257 needs editing to become more understandable.
(p) Towards the end of the Results part, the text is becoming very unclear in many parts:
-line 277: What is meant by a “disruption in several signaling pathways”?
-line 287: “The protein Ubiquitin was found to have the highest degree…” Please explain “highest degree” (of what?)
-line305: “We removed the three smaller connected components…” Please remind the reader of what those components are, or refer to figure.
-line 354-55 “Figure 8 depicts….” Please better explain the “REL activation by PKC through BCL10-MALT1-IKK”. It is difficult to extract the thoughts behind this statement from the figure only.

(q) It is strongly recommended that the parts below are moved to the Discussion. I would also recommend a thorough edition of the text in the results part in general. Many sentences and statements are left without explanations.
Move to discussion:
- Lines 201-207
- 214-223
- 294-304
- 355-379

(r) Discussion:
- It is recommended that the authors discuss and speculate about the reason for downregulation of signaling pathways in relation to the results.
- lines 428-430: ”The examination of the enriched GOBP terms of the network showed an involvement of T cell receptor signaling pathway, B cell receptor signaling pathway, Fc-ε receptor signaling pathway and NF-κB signaling pathway in the differentiation of osteoclasts in the RA synovium.” The involvement of for instance T-cell receptor signaling in osteoclast differentiation is somewhat challenging and needs to be explained/discussed. It is difficult to understand whether the results are referred to the synovia or to osteoclasts in different parts of the mansucript. Similarly, from lines 454-456 it is important to discuss how PRKCZ is involved in osteoclast differentiation (through T-cell signaling?)
-please discuss strengths and weaknesses of the strategy used in the presented study. In addition, would a discussion on the possibility that drug treatment is influencing the reported gene expression be important and interesting. Would it be possible to group data from patients on different treatments?

(s) The conclusions need to be stated in a more concise way.

Reviewer 3 ·

Basic reporting

The article is well written and the language is clear. However, the description of the PPI network (lines 283-397) is too descriptive and difficult to follow. Genes and connections are described one after the other without much context. It will be better to simplify these sections and discuss examples most relevant to the current study.

Fig. 6 is too complex and aspects of it are unnecessary. It will be good to simplify it to only keep the parts that are discussed in the paper.

Experimental design

The methods related to GO enrichment analysis in PPI network is unclear (lines 160-170). Did the authors use GO terms corresponding to all the proteins in PPI for enrichment analysis?

Authors combined GO terms to build custom set for enrichment analysis (lines163, 166). It should be evaluated that this did not induces a bias in the analysis, e.g. using permutation analysis with appropriate random sets or other similar methods.

The pathway level analysis using KEGG to identify up/down-regulated pathways is based solely on the enrichment of the genes. Enrichment of upregulated genes in a pathway does not imply that the pathway output is also up, and vice-versa, without taking the information on repression/activation. Have this been taken into account? It should be clearly assessed if the pathway output is up/down.

Validity of the findings

The findings of this meta-analysis will be valuable for the readers in the field. However, it is not clear to me what is achieved using the PPI network that is not possible otherwise. Maybe the message is diluted in the verbose nature of the PPI description. Focussed writing and highlighting the novel findings in this analysis will certainly help.

Additional comments

No comments.

Reviewer 4 ·

Basic reporting

no comments

Experimental design

no comments

Validity of the findings

no comments

Additional comments

Comments for Shilpa et.al:
In present manuscript authors have utilized published microarray data of synovial cells from patients with RA. Authors curated the available database of GEO, KEGG and STRING compared with enriched genes from the RA microarray studies. With their analysis they come up with canonical NF-κB activation and osteoclastogenesis effected by NCF complex through ROS as a major pathological pathways in RA synovium. Study seems suitable for PeerJ readers and provides interesting tools to study available large public database for RA as well as various other conditions.
Following are my major concerns:
1. Table 1 should include further details about published 7 studies which are the base for the current manuscript. Please include the cell type used, age distribution, cohort, gender/distribution, and most importantly major out-come from the microarray data as they have proposed. Later emphasis should be given that how Shilpa et.al analysis have novel findings compared to these public databases.
2. Further these points should be discussed during the explanation of discrepancies when there is minimal overlap of dysregulated genes in these studies.
3. It is surprising to me that there are no error bars for number of genes dysregulated atleast in one study across the 7 studies and so on.
4. Statistical test for overlapping genes across the studies will be informative.
5. The final results should be tested for how many interactions they hold true for atleast 3 studies.
6. Line 186 about heterogeneity of disease should be reconsidered.
7. Lines 224 -225, further references regarding osteoclasts role in RA pathology is essential.
8. Graphical representation of osteoclast markers should be given across the studies compared to controls.
9. Figure5 explain the meaning of size of nodes for pathways shown.
10. Graphically represent the data for the gene involved in osteoclast differentiation among various studies to support the point and explain if there is any variation between them.
11. Line 307 what is ODP network mean.
12. Which figure have details about OPD network. Showing core and shell network.
13. Line 319 NFKB should be written NF-κB, just like line 332 and I appreciate that author’s have used kappa in line 332.
14. Figure 8 title say activation of REL; however the network shown have many NF-κB family members, so update it accordingly.
15. Why Rel was selected over other family members of NF-κB, line 348 suggests just overexpression. However, many GWAS studies also link Rel with RA. It will be interesting if authors can differentiate Rel specific pathways compare to other NF-κB family members.
16. For figures 8 and 9 provide some weightage (score) to the nodes depending on the hits from the studies utilized to generate network.
17. Suprisingly figure 9 doesn’t have any NF-κB association.

---

## Round 0.2 · Minor Revisions

Please make final minor revisions as requested by reviewer 3 and also provide abbreviation list as requested by reviewer 1.

Reviewer 2 ·

Basic reporting

No comment

Experimental design

No comment

Validity of the findings

No comment

Additional comments

From my point of view the authors have satisfactorily discussed and executed the criticism and suggestions from the first review and I have no further comments. One technical point is that, still, I think that the abbreviation list for availability reasons should be included in the main manuscript if that is in accordance with the journals' instructions.

Reviewer 3 ·

Basic reporting

The flow of the manuscript is improved in the revised version. English is clear and professional.

Experimental design

More experimental details are provided which help understand the methods.

Validity of the findings

No comments.

Additional comments

The legend of Figure 7 should only include up pathways/DEGs, and for Figure 8 should only have down. It's confusing to have both up and down in legends when one of them is not part of the figure.

Regarding the description of pathway enrichment, I am only concerned about the language used to describe the enrichment results (line 447-476). Why not simply say pathways enriched in up/down regulated genes? I feel that's a better description of results rather than up/down pathways, given the pathway output cannot be assessed. However, this is a minor point.

Reviewer 4 ·

Basic reporting

Please see comments to author.

Experimental design

Please see comments to author.

Validity of the findings

Please see comments to author.

Additional comments

Shilpa et. al. have updated all the concerns raised by me during previous revision satisfactorily. Manuscript can be accepted for the PeerJ publication. I indeed like to make to point the authors have explained their points very well.

---

## Round 0.3 · accepted · Accept

Authors have satisfactorily responded to Reviewers all the concerns. Now the manuscript is improved substantially compared to the first version.

#